# Covid-19 and cardiovascular disease in a total population-study of long-term effects, social factors and Covid-19-vaccination

Malin Spetz [1,2] ✉, Yvonne Natt och Dag[1,3], Huiqi Li [1], Fredrik Nyberg [1] & Maria Rosvall[1,3]

Understanding more about the risk of cardiovascular disease in the large population-group with mild Covid-19 is essential since the preventive need might be extensive. This study examined the risk of cardiovascular disease following Covid-19, considering risk periods and prognostic factors i.e., social factors, Covid-19-vaccination and comorbidities. The study cohort included the Swedish population aged 40-75 years (n = 4,095,414). Covid-19 was associated with elevated hazard ratios for all outcomes; ranging from 1.22 (95% confidence interval:1.14-1.31) for acute myocardial infarction to 4.31 (95% confidence interval:4.09-4.55) for pulmonary embolism. The increased risk was most evident among hospitalised individuals, however, also individuals with mild Covid-19 had an elevated risk. Finally, our findings demonstrated increased long-term cardiovascular risk and generally stronger effects of Covid-19 in more vulnerable social groups. In this work, we demonstrate an increased risk of cardiovascular disease after Covid-19, also among mild cases, findings relevant from both a public health and healthcare perspective.

Covid-19 (coronavirus disease 2019), caused by SARS-CoV-2 (severe acute respiratory syndrome coronavirus 2), has caused a worldwide health crisis, and over 7 million individuals are documented to have died due to the infection[1]. Since the start of the Covid-19 pandemic, multiple SARS-CoV-2 variants and sub-variants have evolved, and continue to do so[1], paving way for the seasonal occurrence of Covid-19 as well as for potential future outbreaks. Covid-19 primarily affects the lungs, but even in the early pandemic, there were indications that other organs, such as the cardiovascular system, were susceptible to the disease[2,3]. Cardiovascular diseases (CVD) encompass a variety of disorders affecting the heart and the blood vessels, e.g., coronary heart disease, cerebrovascular disease, deep venous thrombosis (DVT) and pulmonary embolism (PE)[4]. Several potential mechanisms have been discussed to explain the link between Covid-19 and CVD, e.g., inflammatory processes, and cardiac injury involving angiotensin converting enzyme 2 (ACE2) receptors in the heart[3,5]. Various pathophysiological pathways have also been considered regarding the thromboembolic complications associated with Covid-19, such as Covid-19-induced endotheliopathy, platelet activation and hypercoagulability[5].

Many studies regarding cardiovascular complications of Covid-19 include hospitalised individuals with more severe disease[6–8], focus on Covid-19 in the early phases of the pandemic[6–13] and/or have a relatively short follow-up period[6,8,11,13]. However, it is essential to learn more about the risk of cardiovascular disease in the large population group with mild Covid-19, since the preventive need then might be extensive[14]. Currently, there are few population-based studies addressing incident cardiovascular disease after mild Covid-19 (often defined as infection not requiring hospitalisation)[9,10,14–22]; some of these studies were based on UK Biobank data[10,14,15,20], included US Veterans[11,17] or were based on a large UK primary care database[21]. The previous studies have, to some extent, examined when, in the aftermath of Covid-19, the shorter-term risk for CVD is increased, but only

[1]School of Public Health and Community Medicine, Institute of Medicine, Sahlgrenska Academy, University of Gothenburg, Gothenburg, Sweden. [2]Department of strategic healthcare development, Head office, Region Västra Götaland, Gothenburg, Sweden. [3]Department of Social Medicine, FoUUI, Regionhälsan, Region Västra Götaland, Gothenburg, Sweden. ✉e-mail: malin.spetz@gu.se

some of the studies specify the time period during which the long-term risk for events is increased[9,16,18,21,22]. Thus, more research is needed regarding the time-window between Covid-19 and incident CVD during a long observational follow-up. Furthermore, findings have shown a protective effect of Covid-19 vaccination on CVD[23]; however, few of the mentioned population-based studies of cardiovascular risk after Covid-19 have taken Covid-19 vaccination into consideration[17,18,20,21]. Finally, additional readily available prognostic factors previously not much investigated, such as sociodemographic determinants, are needed to assess individuals at potential high risk for suffering from CVD after Covid-19.

In this work, we use linked register data to examine the short- and long-term risk of incident cardiovascular disease and mortality following Covid-19 in a total population cohort from Sweden. In particular, risk periods for cardiovascular outcomes after a SARS-CoV-2 infection are studied. Prognostic factors and heterogeneity of effects due to, e.g., disease severity and social groups are also investigated.

## Results

### Description of the study population
Our study population included 4,095,414 individuals aged 40-75 years (Table 1). Individuals who had ever been infected by SARS-CoV-2 at the end of follow-up (12.9%) were, in comparison to those without a known infection by that time, more often of younger age, women, born in a low- and middle-income country (LMIC), had higher educational level and higher income at baseline, and were somewhat less often still unvaccinated against Covid-19 at end of follow-up. Baseline respiratory disease and obesity were more common among individuals who had been infected by SARS-CoV-2, whereas comorbidities in terms of prior cancer, psychiatric disease, diabetes, kidney failure, inflammatory polyarthritis, coagulation disease ($p = 0.105$), and thyroid disease were more common in the group with no Covid-19. During the study period, 32,355 individuals, (6.1% of those infected with SARS-CoV-2), were classified as having had severe Covid-19 in their first infection.

### Incident cardiovascular disease and mortality after Covid-19
Covid-19 was associated with higher hazard ratios (HRs) for all studied cardiovascular outcomes (Table 2, Fig. 1). There was in total approximately 8 million (range 8.09–8.12 million) person-years of follow-up for all participants. In Model 1 (adjusted for age), the HRs ranged from 1.25 (95% confidence interval (CI) 1.17–1.34) for acute myocardial infarction to 4.34 (95% CI 4.12–4.57) for PE, in the Covid-19 group compared to the not infected group. The elevated HRs persisted also in Model 2 (additionally adjusted for sex, country of birth, income, and education) and Model 3 (further additionally adjusted for comorbidities, and vaccination against Covid-19 i.e., fully adjusted model), with quite limited impact of additional adjustments, e.g., in Model 3 the HRs for acute myocardial infarction turned to 1.22 (95% CI 1.14–1.31) and for PE 4.31 (95% CI 4.09–4.55), respectively. Sensitivity analyses regarding incident cardiovascular outcomes also including data on out-of-hospital deaths from the National Cause of Death Register (NCDR) showed nearly identical findings (Supplementary table 1). Interaction analyses by Covid-19 and vaccination showed persistently higher HRs for all cardiovascular outcomes among individuals with Covid-19 who were non-vaccinated compared to those without Covid-19 who were non-vaccinated. Regarding acute myocardial infarction and heart failure (HF), the HRs were increased also among individuals with Covid-19 who were vaccinated with one dose of a Covid-19 vaccine and for ischaemic heart disease (IHD), DVT and PE, there were increased HRs also among those having received two doses of a Covid-19 vaccine, however, with consistently lower risks compared to those non-vaccinated (Supplementary Fig. 1). Similar patterns of associations, as found for cardiovascular outcomes (described in Table 2, Fig. 1), were seen also for all cause mortality, CVD death and for IHD death in all three models (Supplementary Table 2).

Sensitivity analyses performed among those without prior comorbidities showed corresponding patterns of associations with cardiovascular (Supplementary Table 3) and mortality outcomes (Supplementary Table 4), respectively. Furthermore, sensitivity analyses on the age group 18–39 years showed increased risks for some of the cardiovascular outcomes (i.e., HF and thromboembolic disease; Supplementary Table 5), and mortality (i.e., all cause mortality and cardiovascular mortality; Supplementary Table 6), after a SARS-CoV-2 infection. In addition, sensitivity analyses including only older individuals (>75 years) showed increased risks for most of the studied cardiovascular outcomes (Supplementary Table 7), and all mortality outcomes (Supplementary Table 8) after Covid-19. Moreover, sensitivity analysis beginning follow-up on 1 March 2020 showed nearly identical results both with regard to incident cardiovascular disease and mortality (Supplementary Table 9). Finally, sensitivity analyses excluding those who immigrated during the five-year lookback period did not change the initial patterns of associations seen in Table 2 and Supplementary Table 2 (Supplementary Table 10).

### Cardiovascular risk and mortality by Covid-19 severity
Analyses by severity of Covid-19 showed that HRs for all outcomes were higher for severe Covid-19 (Fig. 2, Supplementary Table 11). Regarding severe Covid-19, the risks were substantially elevated, most notably for PE (HR 22.94 [95% CI 21.52–24.44] in Model 3), followed by DVT, intracerebral haemorrhage, cardiomyopathy, and HF with HRs between 4 and 5. Among those with mild disease, there were increased risks for IHD (HR 1.07 [95% CI 1.01–1.13]) as well as for thromboembolic disease, with HRs for DVT and PE of 1.41 (95% CI 1.31–1.52) and 1.78 (95% CI 1.64–1.92) in Model 3, respectively. Furthermore, there were substantially increased risks for all cause mortality, CVD death and IHD death among those hospitalised due to Covid-19, while in the group with mild disease, there was a much more discrete increased risk of death from any cause and overall CVD mortality, but not from IHD (Supplementary Table 12). For example, in Model 3, the HR for all cause mortality was 10.77 (95% CI 10.33–11.24) in the hospitalised group, whereas it was 1.49 (95% CI 1.42–1.57) in the group with mild disease.

Stratified analyses based on pandemic waves showed similar patterns of associations with incident cardiovascular disease across periods, however, the HRs were generally more elevated during the first and second pandemic waves, and least elevated during the fourth pandemic wave (Supplementary Table 13). Regarding mortality outcomes, there was a successive reduction in risk over time, with the highest risk during the first pandemic wave. In addition, stratified analyses based on predominant virus variants showed the most elevated risk for cardiovascular disease and mortality during the time period dominated by the mixed variants and the alpha variant, especially for mortality outcomes (Supplementary Table 14).

### Risk of cardiovascular disease and mortality by risk periods
The risk following Covid-19 was highest for all outcomes in the first 0–14 days after an infection (Fig. 3, Supplementary Table 15). For example, the HR for PE was 44.15 (95% CI 41.12–47.40) during the first two weeks following Covid-19 and decreased successively to no risk in the last time window 366–730 days after infection. During the time-window 91–180 days after infection, the risk was still slightly increased for acute myocardial infarction, IHD, HF, DVT, PE, all cause mortality and cardiovascular disease mortality (statistically significant HR point estimates 1.20 to 1.30). For ischaemic stroke, cerebrovascular disease, HF, DVT, all cause mortality and cardiovascular disease mortality, the increased risk remained even longer (366–730 days).

### Risk of three key cardiovascular outcomes by social factors
Analyses for the three a priori selected broader key outcomes; cerebrovascular disease, IHD and thromboembolic disease

**Table 1 | Description of the study population**

| | Ever infected by Covid-19 at end of follow-up | | p-value | Total |
|---|---|---|---|---|
| | **No** | **Yes** | | |
| N | 3,568,017 (87.1%) | 527,397 (12.9%) | | 4,095,414 (100.0%) |
| Mean age with SD (1 January 2020) | 56.8 (10.2) | 52.2 (8.6) | <0.001[f] | 56.2 (10.2) |
| Age group (years) | | | | |
| 40–54 | 1,482,102 (41.5%) | 319,201 (60.5%) | <0.001[g] | 1,801,303 (44.0%) |
| 55–64 | 999,900 (28.0%) | 146,698 (27.8%) | | 1,146,598 (28.0%) |
| 65-75 | 1,086,015 (30.4%) | 61,498 (11.7%) | | 1,147,513 (28.0%) |
| Sex | | | | |
| Men | 1,760,314 (49.3%) | 255,747 (48.5%) | <0.001[g] | 2,016,061 (49.2%) |
| Women | 1,807,703 (50.7%) | 271,650 (51.5%) | | 2,079,353 (50.8%) |
| Country of birth[a] | | | | |
| Sweden | 2,846,811 (79.8%) | 396,588 (75.2%) | <0.001[g] | 3,243,399 (79.2%) |
| HIC | 287,996 (8.1%) | 35,787 (6.8%) | | 323,783 (7.9%) |
| LMIC | 432,724 (12.1%) | 94,960 (18.0%) | | 527,684 (12.9%) |
| Income[b] | | | | |
| Low | 1,221,979 (34.2%) | 142,555 (27.0%) | <0.001[g] | 1,364,534 (33.3%) |
| Medium | 1,170,984 (32.8%) | 193,754 (36.7%) | | 1,364,738 (33.3%) |
| High | 1,173,641 (32.9%) | 191,005 (36.2%) | | 1,364,646 (33.3%) |
| Education[c] | | | | |
| Primary | 534,368 (15.0%) | 61,055 (11.6%) | <0.001[g] | 595,423 (14.5%) |
| Secondary | 1,600,177 (44.8%) | 234,381 (44.4%) | | 1,834,558 (44.8%) |
| Tertiary | 1,381,328 (38.7%) | 227,056 (43.1%) | | 1,608,384 (39.3%) |
| Comorbidities (2015–2019) | | | | |
| Respiratory disease | 279,379 (7.8%) | 43,817 (8.3%) | <0.001[g] | 323,196 (7.9%) |
| Cancer | 206,271 (5.8%) | 20,749 (3.9%) | <0.001[g] | 227,020 (5.5%) |
| Psychiatric disease | 131,659 (3.7%) | 15,936 (3.0%) | <0.001[g] | 147,595 (3.6%) |
| Diabetes | 127,318 (3.6%) | 15,327 (2.9%) | <0.001[g] | 142,645 (3.5%) |
| Kidney failure | 23,903 (0.7%) | 2954 (0.6%) | <0.001[g] | 26,857 (0.7%) |
| Inflammatory polyarthritis | 66,313 (1.9%) | 8502 (1.6%) | <0.001[g] | 74,815 (1.8%) |
| Thyroidal disease | 77,896 (2.2%) | 11,275 (2.1%) | 0.035[g] | 89,171 (2.2%) |
| Coagulation disease | 11,213 (0.3%) | 1587 (0.3%) | 0.105[g] | 12,800 (0.3%) |
| Obesity | 51,535 (1.4%) | 9213 (1.7%) | <0.001[g] | 60,748 (1.5%) |
| Vaccine doses at the end of follow-up[d] | | | | |
| 0 | 361,676 (10.1%) | 53,238 (10.0%) | | 414,914 (10.1%) |
| 1 | 47,384 (1.3%) | 11,883 (2.3%) | <0.001[g] | 59,267 (1.4%) |
| ≥2 | 3,158,957 (88.5%) | 462,276 (87.7%) | | 3,621,233 (88.4%) |
| Hospitalisation due to Covid-19[e] | 0 (0.0%) | 32,355 (6.1%) | <0.001[g] | 32,355 (0.8%) |

[a]Country of birth: Sweden, HIC: High Income Countries; LMIC: Low- and middle Income Countries.
[b]Income: The disposable income per consumption unit; High: 3rd tertile, Medium: 2nd tertile Low: 1st tertile.
[c]Education: primary schooling (<10 years); secondary schooling (10–12 years); tertiary schooling (>12 years).
[d]Vaccinated with a Covid-19 vaccine, time-varying variable.
[e]Hospitalisation with a Covid-19 diagnosis within 14 days after a first positive PCR-test or 2 days before the positive PCR-test for SARS-CoV-2.
[f]Comparison was performed using a two-sided t-test.
[g]Two-sided Chi-2 tests were performed.
Distribution of sociodemographic factors, comorbidities, vaccination and hospitalisation due to Covid-19 among individuals ever infected by Covid-19 at the end of follow-up, i.e., 31 December 2021 (no/yes) in the Swedish population aged 40–75, presented as percentages.

(DVT or PE), stratified by social factors i.e., sex, age, education, income and country of birth, indicated that the associations between Covid-19 and CVD outcomes were more pronounced in the strata of men, high age, low income, and low education (Table 3). However, the pattern was less consistent with regard to country of birth. Sensitivity analyses performed among those without prior comorbidities showed similar patterns of associations with cardiovascular outcomes (Supplementary Table 16) as did sensitivity analyses excluding those who immigrated during the five-year lookback period (Supplementary Table 17).

## Discussion

In our study, encompassing the total population of Sweden ages 40–75 years, Covid-19 was associated with both incident CVD and mortality, even after adjustments for vaccination, co-morbidities and socio-demographic factors. Although the cardiovascular risk was highest during the two weeks after Covid-19, the increased risk remained for some outcomes more than one year after a SARS-CoV-2 infection. Analyses by severity of Covid-19 showed a consistent increased risk for all studied outcomes among those hospitalised due to Covid-19, while non-hospitalised individuals had an increased risk for IHD, thromboembolic disease and mortality.

**Table 2 | Risk of cardiovascular disease following Covid-19**

| Incident CVD outcomes | Covid-19[c] | Model 1[d]<br>HR (95% CI) | Model 2[f]<br>HR (95% CI) | Model 3[g]<br>HR (95% CI) |
|---|---|---|---|---|
| **Ischaemic stroke** | | | | |
| (n[a] = 15 175; IR[b] = 187.1 (95% CI:184.2–190.1)) | No | 1.00[e] | 1.00[e] | 1.00[e] |
| | Yes | 1.37 (1.27–1.48) | 1.41 (1.30–1.52) | 1.38 (1.27–1.48) |
| **Intracerebral haemorrhage** | | | | |
| (n[a] = 3 092; IR[b] = 38.1 (95% CI:36.8–39.4)) | No | 1.00[e] | 1.00[e] | 1.00[e] |
| | Yes | 1.42 (1.21–1.66) | 1.47 (1.25–1.72) | 1.44 (1.23–1.69) |
| **Cerebrovascular disease** | | | | |
| (n[a] = 25 117; IR[b] = 310.1 (95% CI:306.2–313.9)) | No | 1.00[e] | 1.00[e] | 1.00[e] |
| | Yes | 1.43 (1.35–1.51) | 1.48 (1.39–1.57) | 1.45 (1.36–1.53) |
| **Acute myocardial infarction** | | | | |
| (n[a] = 17 589; IR[b] = 217.0 (95% CI:213.8–220.2)) | No | 1.00[e] | 1.00[e] | 1.00[e] |
| | Yes | 1.25 (1.17–1.34) | 1.24 (1.16–1.33) | 1.22 (1.14–1.31) |
| **Ischaemic heart disease** | | | | |
| (n[a] = 39 665; IR[b] = 490.6 (95% CI:485.8–495.4)) | No | 1.00[e] | 1.00[e] | 1.00[e] |
| | Yes | 1.40 (1.34–1.47) | 1.37 (1.31–1.44) | 1.35 (1.28–1.41) |
| **Cardiomyopathy** | | | | |
| (n[a] = 3 431; IR[b] = 42.3 (95% CI:40.9–43.7)) | No | 1.00[e] | 1.00[e] | 1.00[e] |
| | Yes | 1.36 (1.18–1.57) | 1.41 (1.22–1.63) | 1.39 (1.21–1.61) |
| **Heart failure** | | | | |
| (n[a] = 20 716; IR[b] = 255.6 (95% CI:252.1-259.1)) | No | 1.00[e] | 1.00[e] | 1.00[e] |
| | Yes | 1.56 (1.47-1.66) | 1.63 (1.53-1.74) | 1.54 (1.45-1.64) |
| **Deep venous thrombosis** | | | | |
| (n[a] = 14 135; IR[b] = 174.3 (95% CI:171.4–177.2)) | No | 1.00[e] | 1.00[e] | 1.00[e] |
| | Yes | 1.73 (1.63–1.85) | 1.80 (1.68–1.92) | 1.77 (1.66–1.89) |
| **Pulmonary embolism** | | | | |
| (n[a] = 13 080; IR[b] = 161.2 (95% CI:158.5–164.0)) | No | 1.00[e] | 1.00[e] | 1.00[e] |
| | Yes | 4.34 (4.12–4.57) | 4.46 (4.23–4.70) | 4.31 (4.09–4.55) |

[a]Number of events during follow-up.
[b]IR: Incidence rate per 100,000 person-years.
[c]Defined as a time-varying exposure.
[d]Model 1: Adjusted for age.
[e]Reference group.
[f]Model 2: Adjusted for age, sex, country of birth, income and education.
[g]Model 3: Adjusted for age, sex, country of birth, income, education, comorbidities and vaccination against Covid-19.
Risk of various cardiovascular disease (CVD) outcomes related to Covid-19 [hazard ratios (HRs) with 95% confidence intervals (CI)] in a cohort of individuals in Sweden aged 40–75 years (n = 4,095,414) followed from 1 January 2020 to 31 December 2021, from three different Cox regression models with different adjustments.

The large dataset used in our study enabled analyses of multiple CVD diagnose groups. The results showed, in line with previous studies, increased risks of IHD[9,13–15,17,21], cerebrovascular disease[9,13–15,17,19], HF[9,14,15,17], cardiomyopathy[17], thromboembolic disease[9,12,14–17,20–22], and mortality[10,11,14,15], following Covid-19. The mentioned population-based studies have used varying approaches to investigate the time course for cardiovascular disease risk after Covid-19[9,10,13–19,21,22], however, only some of these studies have examined and specified the time-period for increased long-term risk of cardiovascular complications[9,16,18,21,22]. In the Swedish studies by Katsoularis et al., it was demonstrated, both in the study of DVT, PE, and bleeding[16] and in the study of arrhythmias[18] after Covid-19, that the risk of events varied between specific time windows up to 180 days after infection. For example, the risk for PE following Covid-19 was highest during the first two weeks after an infection, showing incidence rate ratios of 36.17 (95% CI 31.55–41.47) for the first week and 46.40 (95% CI 40.61–53.02) for the second week, respectively, and the risk remained increased up to 110 days after infection[16]. Furthermore, Sjöland et al. showed an increased risk of venous thromboembolism persisting after 180 days among those hospitalised for Covid-19, while there was no increased long-term risk among individuals not hospitalised[22].

In a study based on a large UK primary care database, increased risks for venous thromboembolism and atherosclerotic cardiovascular events were seen up to 60 days following infection (follow-up time 180 days)[21]. Furthermore, a Danish national cohort study with a focus on incident stroke after Covid-19 demonstrated an increased risk of stroke in the acute phase after Covid-19, and the increased risk persisted for hospitalised individuals also during the post-infection period; however, there was no increased long-term risk of stroke among individuals with community-managed Covid-19[19]. A study including US veterans had a follow-up period up to one year, but did not define risk windows for cardiovascular events after Covid-19[17]. Thus, our study aligns with previous findings demonstrating that the risk for cardiovascular complications is highest during the first weeks after an infection, but also adds evidence by examining in more detail the long-term risk for different cardiovascular outcomes during specified time windows following Covid-19. For example, our findings demonstrated a slightly increased risk for acute myocardial infarction, IHD, HF, DVT, PE, all cause mortality, and cardiovascular disease mortality extending into the time-window 91-180 days after infection. For the longest time-window examined (366–730 days), the risks for ischaemic stroke,

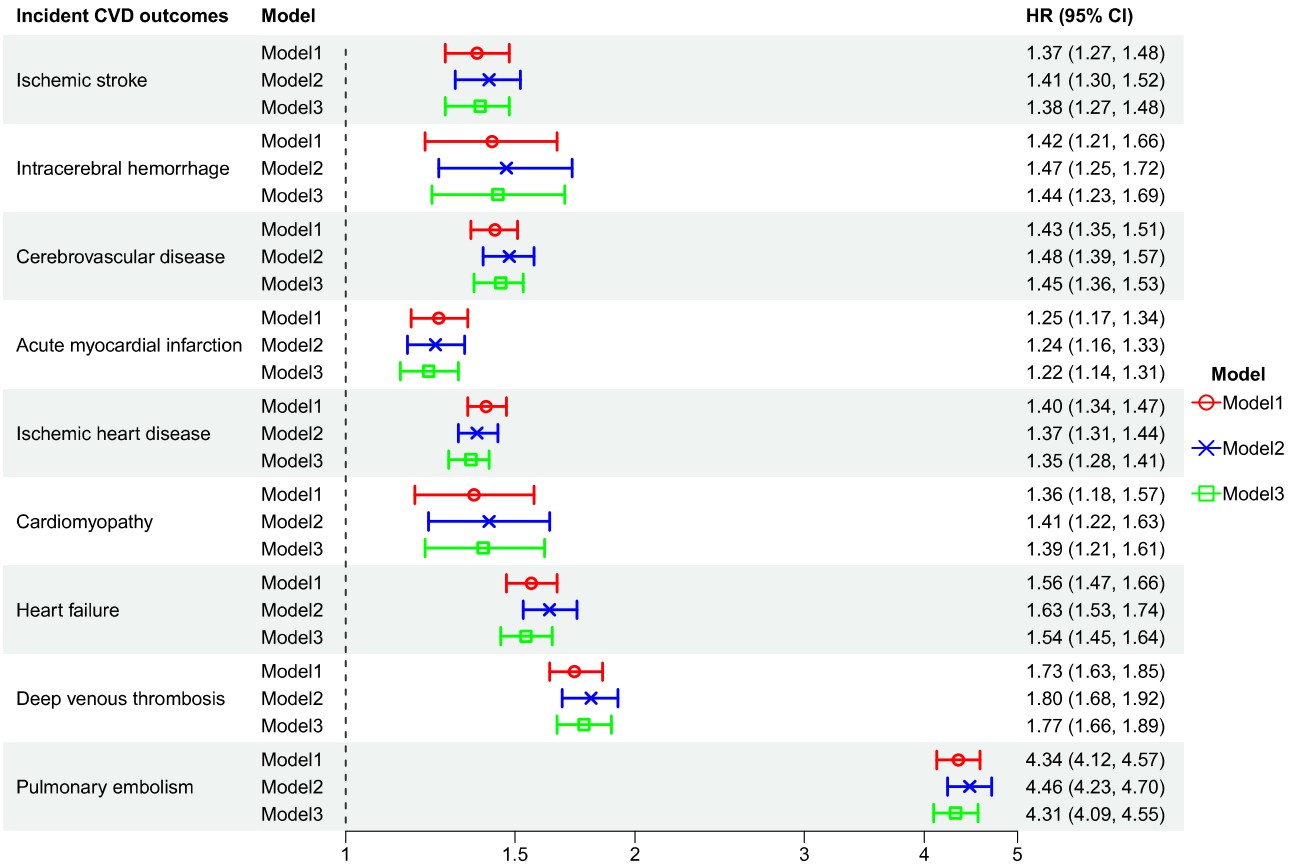

**Fig. 1 | Risk of cardiovascular disease following Covid-19.** Data is presented as hazard ratios (HR) with 95% confidence intervals (CI) of various cardiovascular disease outcomes related to Covid-19 (*n* = 4,095,414). Model 1 (red): Adjusted for age; Model 2 (blue): Adjusted for age, sex, country of birth, income and education; Model 3 (green): Adjusted for age, sex, country of birth, income, education, comorbidities and vaccination against Covid-19.

cerebrovascular disease, HF, DVT, all cause mortality and cardiovascular disease mortality were increased.

Analyses by severity of Covid-19 showed that non-hospitalised individuals had an increased risk for IHD, thromboembolic disease and mortality (all cause mortality and CVD mortality), which is consistent with the results from previous population-based studies. An earlier Swedish study based on the general population demonstrated that mild Covid-19 is a risk factor for DVT and PE with HRs of 2.80 (95% CI 2.26–3.47) and 6.77 (95% CI 5.43–8.45), respectively[16]. Similarly, Sjöland et al. showed that individuals not hospitalised for Covid-19 had an increased risk for venous thromboembolism during a period up to 60 days after infection, however, with no increased long-term risk[22]. Also, in a British study including UK Biobank participants, Covid-19 was associated with a higher risk for venous thromboembolism, HR 2.74 (95% CI 1.38–5.45), and all cause death, HR 10.23 (95% CI 7.63–13.70), among non-hospitalised individuals[14]. Another UK Biobank study likewise demonstrated that Covid-19 increases the risk for major CVD (composite of stroke, coronary heart disease, and HF) and mortality after both severe and non-severe Covid-19, although severe infections were defined as patients with critical care and thus non-severe cases also included hospitalised individuals[15]. Furthermore, a study of US Veterans showed an increased risk for several cardiovascular outcomes among individuals alive 30 days after a first positive Covid-19, also after mild (non-hospitalised) disease, e.g., HR 2.01 (95% CI 1.84–2.19) for PE, and HR 1.62 (95% CI 1.49–1.76) for DVT[17]. In addition, findings from a study based on a large UK primary care database demonstrated an increased risk of thromboembolic events following Covid-19[21]. Also, in a UK Biobank study, high venous thromboembolic risk was observed among ambulatory Covid-19 patients (HR 21.42; 95% CI, 12.63–36.31)[20]. Thus, our study in a total population cohort strengthens the previously demonstrated increased risk for thromboembolic complications, IHD, and mortality, also after mild Covid-19.

In addition to the severity of Covid-19 as a prognostic factor for cardiovascular outcomes, we also studied the impact of social determinants. Our findings indicated stronger associations between Covid-19 and CVD outcomes among individuals with male sex, high age, low income, and low education. These groups are the commonly recognised vulnerable groups for Covid-19 itself and Covid-19-related outcomes such as persistent Covid-19 symptoms[24] and sick leave[25]. Some earlier studies have investigated social determinants as prognostic factors. For example, the previously mentioned Swedish study showed that older age had a significant effect on the association between Covid-19 and DVT and PE, while male sex had a significant effect only on the association between Covid-19 and PE[16]. However, in another study by the same group, using a similar methodology, there was no effect modification by age nor by sex on myocardial infarction or stroke[13].

Previous studies have demonstrated that Covid-19 vaccines prevent against cardiovascular disease after Covid-19[23]. Furthermore, Xu et al. demonstrated, also using data from SCIFI-PEARL, that the risk for various cardiac outcomes, among those aged >40 years, decreased after Covid-19 vaccination, even though there was a transiently increased risk for extrasystoles. Moreover, there was no difference in cardiac risk reduction between vaccine products (i.e., BNT162b2 (Pfizer-BioNTech), mRNA-1273 (Moderna), or AZD1222 (AstraZeneca))[26]. However, few of the population-based studies on

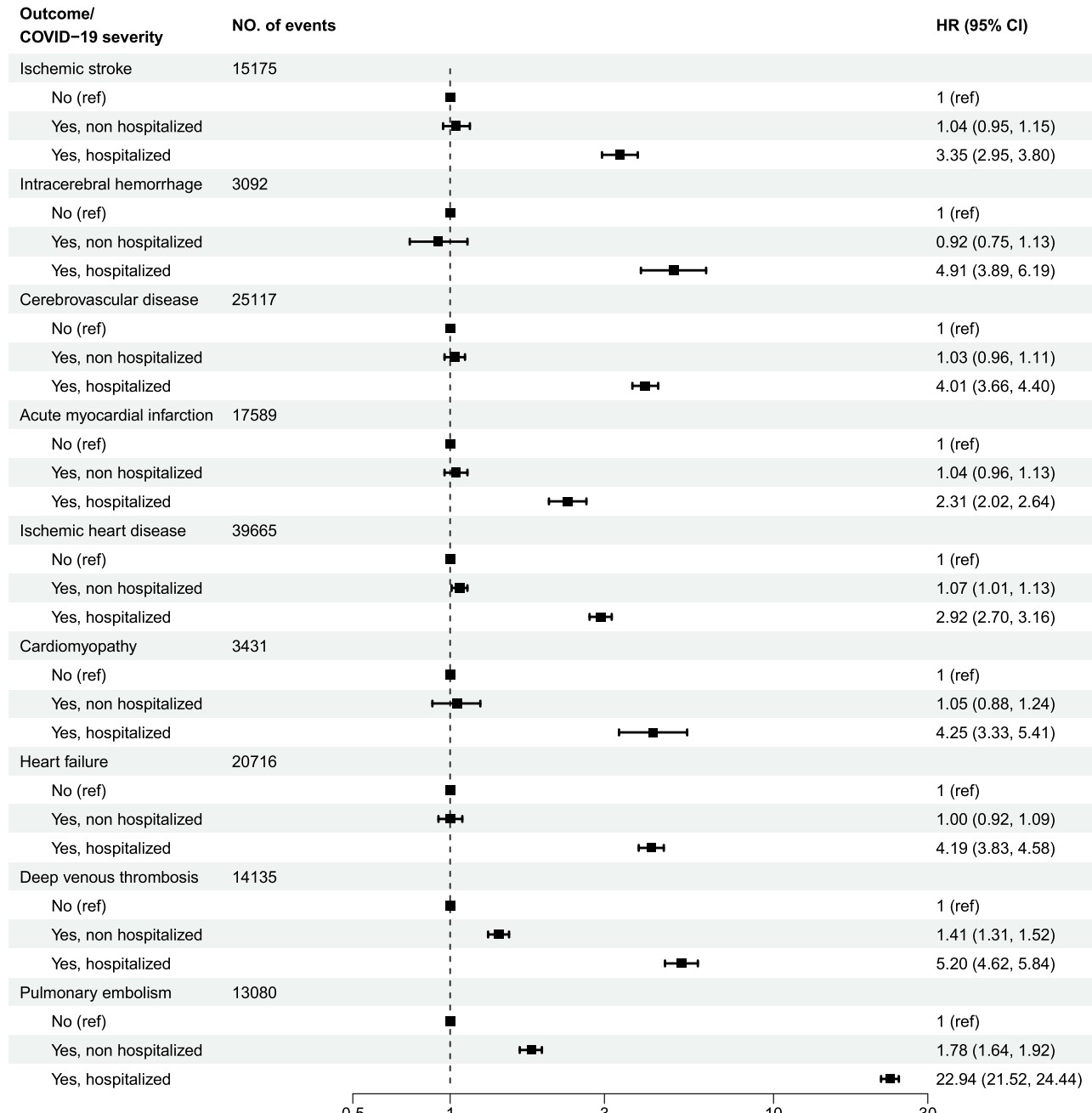

**Fig. 2 | Risk of cardiovascular disease following Covid-19 by severity.** Data is presented as hazard ratios (HR) with 95% confidence intervals (CI) of various cardiovascular disease outcomes related to Covid-19 (*n* = 4,095,414), by severity (i.e., hospitalisation or not) with multiple adjustments.

risk of cardiovascular disease after Covid-19 have adjusted or stratified for vaccination[17,18,20,21]. For example, in a UK Biobank study, the risk of incident venous thromboembolism among ambulatory Covid-19 patients still persisted; however, it was reduced in those fully vaccinated[20]. In the study based on a large UK primary care database, incidence rates were similar for most cardiovascular events in those vaccinated and unvaccinated, respectively, except for a lower incidence of PE among those who were vaccinated[21]. In our study, Covid-19 vaccination was treated as a time-varying variable in all analyses, and our findings demonstrated increased risks for all studied outcomes after Covid-19, even after adjustment for vaccination. The impact of vaccination was further studied with interaction analysis to assess whether the CVD risk following Covid-19 was modified by vaccination. The results showed an increased risk

for all studied cardiovascular outcomes following Covid-19 among non-vaccinated individuals; however, among those who had received at least one vaccine dose, the heightened risk for most cardiovascular outcomes after an infection was attenuated, indicating a cardioprotective role of the vaccine.

Our study has several strengths. Firstly, the study population includes the total Swedish population aged 40–75 years, which makes the results well-representative and generalisable. Secondly, the Swedish Personal Identity Number (PIN), unique for each individual, makes it possible to link numerous registers with extremely high accuracy and few missing data[27]. Thirdly, the Swedish nationwide registers have documented high quality, and since the reporting of positive SARS-CoV-2 tests is mandatory according to Swedish law, data regarding Covid-19 is very reliable, and testing was strongly

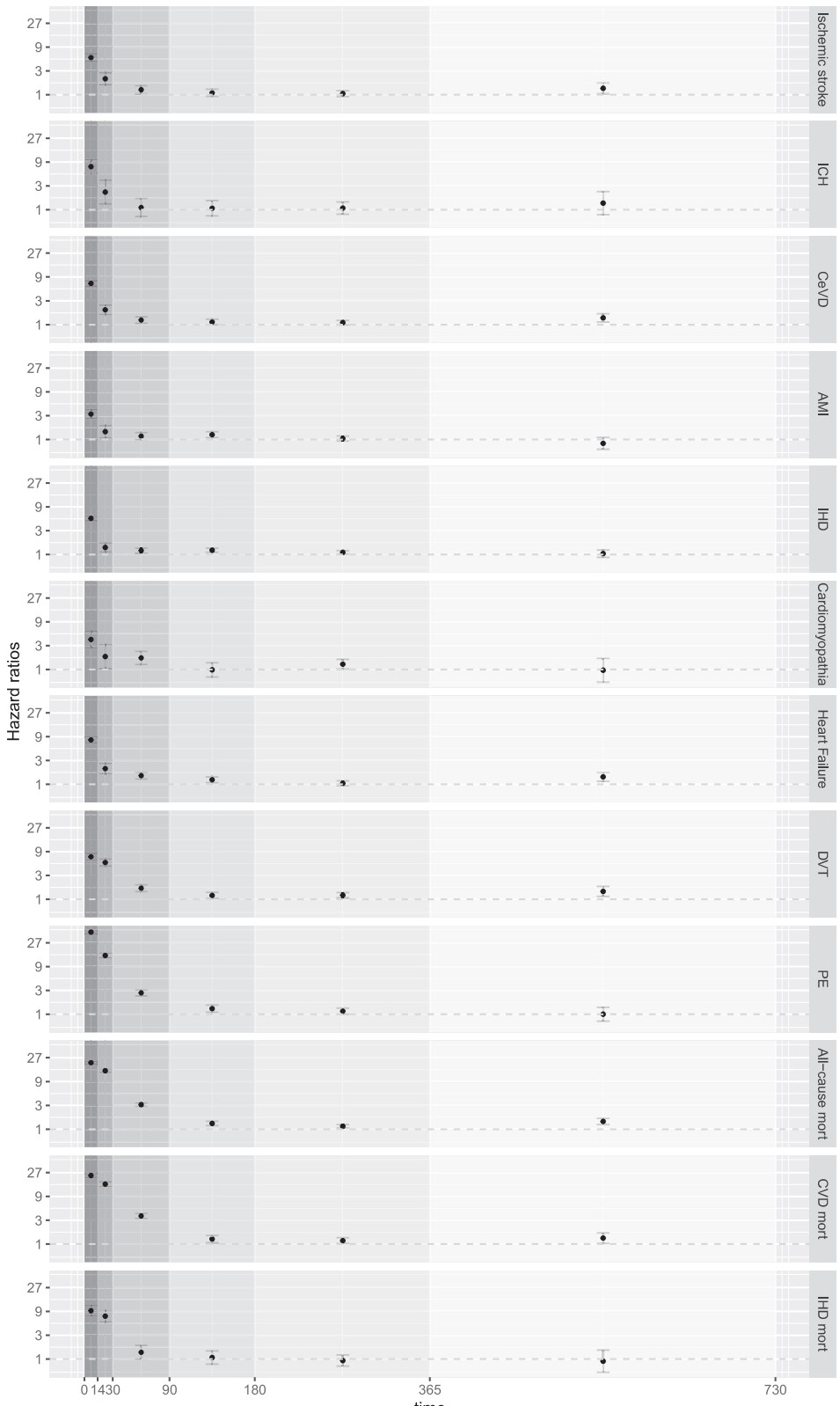

**Fig. 3 | Risk of cardiovascular disease and mortality following Covid-19 by risk periods.** Data is presented as hazard ratios (HR) with 95% confidence intervals (CI) of various cardiovascular disease and mortality outcomes related to Covid-19 (*n* = 4,095,414), by risk periods, compared to the non-infected group with multiple adjustments. Time period after Covid-19 (days): 0–14, 15–30, 31–90, 91–180, 181–365 and 366–730. ICH Intracerebral haemorrhage, CeVD Cerebrovascular disease, AMI Acute myocardial infarction, IHD Ischaemic heart disease, DVT Deep venous thrombosis, PE Pulmonary embolism, CVD Cardiovascular disease. The numerical values of the data shown in the figure are provided in Supplementary Table 15.

**Table 3 | Risk of three major cardiovascular disease-groups following Covid-19 stratified by social factors**

| Exposure strata | | Cardiovascular outcomes | | |
|---|---|---|---|---|
| | | Cerebrovascular disease | Ischaemic heart disease | Thromboembolic disease |
| Covid-19[a] | Sex[b] | HR (95% CI) | HR (95% CI) | HR (95% CI) |
| No | Men | 1·00[i] | 1·00[i] | 1·00[i] |
| No | Women | 0.65 (0.64–0.67) | 0.39 (0.38–0.40) | 0.83 (0.81–0.85) |
| Yes | Men | 1.57 (1.46–1.68) | 1.41 (1.33–1.49) | 3.48 (3.31–3.67) |
| Yes | Women | 0.83 (0.76–0.91) | 0.47 (0.43–0.51) | 1.88 (1.76–2.00) |
| Covid-19[a] | Age[e] | | | |
| No | 40–54 years | 1·00[i] | 1·00[i] | 1·00[i] |
| No | 55–64 years | 2.78 (2.66–2.91) | 3.48 (3.36–3.60) | 1.78 (1.71–1.84) |
| No | 65–75 years | 6.12 (5.88–6.37) | 7.09 (6.86–7.32) | 2.78 (2.69–2.88) |
| Yes | 40–54 years | 1.30 (1.16–1.46) | 1.23 (1.11–1.35) | 2.47 (2.30–2.64) |
| Yes | 55–64 years | 3.65 (3.29–4.05) | 4.11 (3.79–4.45) | 4.95 (4.61–5.31) |
| Yes | 65–75 years | 9.58 (8.72–10.51) | 10.51 (9.75–11.34) | 9.63 (8.95–10.37) |
| Covid-19[a] | Income[d] | | | |
| No | Low | 1·00[i] | 1·00[i] | 1·00[i] |
| No | Medium | 0.75 (0.73–0.78) | 0.93 (0.91–0.96) | 0.94 (0.91–0.97) |
| No | High | 0.68 (0.66–0.70) | 0.88 (0.86–0.90) | 0.89 (0.86–0.93) |
| Yes | Low | 1.57 (1.43–1.72) | 1.46 (1.36–1.58) | 3.31 (3.09–3.55) |
| Yes | Medium | 1.02 (0.92–1.14) | 1.26 (1.16–1.36) | 2.65 (2.47–2.84) |
| Yes | High | 0.94 (0.85–1.04) | 1.07 (0.99–1.17) | 2.37 (2.21–2.54) |
| Covid-19[a] | Education[e] | | | |
| No | Primary | 1·00[i] | 1·00[i] | 1·00[i] |
| No | Secondary | 0.92 (0.89–0.95) | 0.93 (0.90–0.95) | 0.98 (0.94–1.01) |
| No | Tertiary | 0.78 (0.75–0.81) | 0.72 (0.70–0.74) | 0.86 (0.82–0.89) |
| Yes | Primary | 1.56 (1.38–1.77) | 1.48 (1.35–1.63) | 3.41 (3.11–3.75) |
| Yes | Secondary | 1.29 (1.18–1.41) | 1.18 (1.10–1.27) | 2.73 (2.56–2.91) |
| Yes | Tertiary | 1.11 (1.00–1.23) | 0.96 (0.88–1.05) | 2.38 (2.21–2.56) |
| Covid-19[a] | Country of birth[f] | | | |
| No | Sweden | 1·00[i] | 1·00[i] | 1·00[i] |
| No | HIC[g] | 1.02 (0.97–1.07) | 1.19 (1.15–1.24) | 0.87 (0.83–0.92) |
| No | LMIC[h] | 0.86 (0.82–0.90) | 1.52 (1.47–1.57) | 0.62 (0.59–0.66) |
| Yes | Sweden | 1.40 (1.31–1.50) | 1.29 (1.22–1.37) | 2.65 (2.53–2.78) |
| Yes | HIC[g] | 1.61 (1.33–1.95) | 1.70 (1.45–1.99) | 2.79 (2.43–3.20) |
| Yes | LMIC[h] | 1.37 (1.21–1.56) | 2.23 (2.05–2.44) | 2.78 (2.55–3.03) |

[a]Defined as a time-varying exposure.
[b]Adjusted for age, country of birth, income, education, comorbidities and vaccination against Covid-19.
[c]Adjusted for sex, country of birth, income, education, comorbidities and vaccination against Covid-19.
[d]Adjusted for age, sex, country of birth, education, comorbidities and vaccination against Covid-19
[e]Adjusted for age, sex, country of birth, income, comorbidities and vaccination against Covid-19.
[f]Adjusted for age, sex, income, education, comorbidities and vaccination against Covid-19.
[g]HIC = High income countries.
[h]LMIC = Low- and middle-income countries.
[i]Reference group.
Risk of cerebrovascular disease, ischaemic heart disease and thromboembolic disease (deep venous thrombosis and/or pulmonary embolism) respectively, related to Covid-19 [hazard ratios (HRs) with 95% confidence intervals (CI)] in a cohort of individuals in Sweden aged 40–75 years (n = 4,095,414) followed from 1 January 2020 to 31 December 2021, from fully adjusted Cox regression models, and stratified by sex, age, income, education and country of birth.

recommended and widespread during the study period, except for the very beginning of the pandemic. Fourthly, all our analyses were adjusted for relevant sociodemographic factors and comorbidities, and adjustment for Covid-19 vaccination was made using a time-varying variable. A final strength in our study is the detailed information regarding vaccination status (to report Covid-19 vaccination is also mandatory and regulated by Swedish law), and the study period includes the time period when vaccines against Covid-19 became available to the general population.

A limitation in this study is that, during the early pandemic, the test capacity for SARS-CoV-2 was limited, and individuals with mild Covid-19 were not prioritised. However, the restricted testing implying that some individuals with Covid-19 were classified as non-infected, which could rather underestimate the demonstrated risks for CVD. In addition, stratified analyses showed similar patterns across pandemic waves, although with stronger associations during the first and second pandemic waves, especially for mortality outcomes, indicating robustness of findings despite variation in SARS-CoV-2 test capacity. Another possible weakness is that multiple infections with Covid-19 were not considered in our analyses since information on reinfection was limited in the Swedish Covid-19 surveillance[28]. An additional limitation is that in order to interpret a summary HR as valid throughout follow-up, all covariates included in the Cox model are assumed to have more or less proportional effects across time, which may not always be the case. Yet our

wave-specific analysis provided consistent patterns of associations, which potentially indicates that this issue is not a substantial concern. Furthermore, using hospitalisation as a proxy of disease severity could also be a potential limitation, i.e., due to possible variation in the criteria for being admitted to hospital during different pandemic waves. However, stratified analyses based on pandemic wave showed similar patterns of associations with incident cardiovascular disease both among those hospitalised and non-hospitalised, although, as previously mentioned, with the most elevated HRs during the first and second pandemic waves. Moreover, a potential concern was starting follow-up already in a period from the first case when broad community transmission had not yet occurred; however, sensitivity analysis with a later start of follow-up gave identical results to the main analysis. This finding is to be expected, given that the Cox regression method will give very little weight to the early months with few cases in producing the overall HR estimate. A final study limitation is that we have no data regarding confounding factors such as lifestyle; however, a previous Swedish study using two different methodologies showed similar results in self-controlled case series analysis and a matched cohort analysis (not including lifestyle), indicating that such factors might not be major confounders[16]. Lastly, it is important, in a prospective study of the cardiovascular consequences of Covid-19, to bear in mind the changing nature of the pandemic and to consider implications of, for example, new approaches in the treatment of Covid-19, but also the development of new variants of SARS-CoV.

In conclusion, our study, based on the total Swedish population aged 40–75 years, demonstrated that Covid-19 was associated with an increased risk of both incident cardiovascular disease and mortality (i.e., all cause mortality, CVD mortality and IHD mortality). Additionally, our results showed that although the cardiovascular risk following Covid-19 was highest during the first two weeks after an infection, the risk persisted for some cardiovascular outcomes for more than one year after the infection. Individuals hospitalised with Covid-19 had an increased risk for all studied outcomes, but individuals with mild disease also had clear but less strongly elevated risks for IHD, thromboembolic disease, all cause mortality and CVD mortality. Finally, our findings indicate generally stronger effects of Covid-19 on the broader cardiovascular disease outcomes among more vulnerable social groups. Our study thus contributes to the important understanding of the potential long-term impact of the pandemic on cardiovascular health and might help to improve preventive strategies for managing cardiovascular complications following Covid-19.

## Methods

This study complies with relevant ethical regulations, and permissions for the research have been obtained from the Swedish Ethical Review Authority (Approval no. 2020-01800 with subsequent amendments).

### Study design and study population

Data from the SCIFI-PEARL (Swedish COVID-19 Investigation for Future Insights—a Population Epidemiology Approach using Register Linkage) study were used[29]. SCIFI-PEARL is a nationwide observational research project focused on Covid-19, linking numerous Swedish population and health registers using the unique individual Swedish PIN, for the total Swedish population.

The study cohort for the current study included all individuals residing in Sweden on 1 January 2020 (i.e., index date), aged 40–75 years ($n = 4,371,103$). The rationale for the lower age limit was grounded in the low CVD incidence rate among younger individuals, combined with differing pathophysiological mechanisms, where genetic and autoimmune disorders play a relatively more prominent role[30–32]. The higher age limit was based on restricted and delayed test capacity for SARS-CoV-2 among the elderly in the early pandemic[33]. Individuals with prior cardiovascular events corresponding to the studied outcomes during the five years before the index date ($n = 275,689$) were excluded, resulting in a study population of 4,095,414 individuals. On 1

April 2022, changes in the Swedish Communicable Disease Act substantially reduced the volume of polymerase chain reaction (PCR) testing for SARS-CoV-2 in the general population[34]. The end of follow-up was defined as 31 December 2021 to ensure a time margin if, potentially, there was a reduction in PCR testing also during a period preceding the law changes.

### Outcomes

The cardiovascular outcomes were defined based on primary and secondary diagnoses in the International Classification of Diseases, version 10 (ICD-10), in the National Patient Register (NPR), from specialist outpatient visits and hospitalisations. They included ischaemic stroke (ICD-10 codes I63–I64), intracerebral haemorrhage (I61), cerebrovascular disease (I60–I69), acute myocardial infarction (I21), IHD (I20-I25), cardiomyopathy (I42), HF (I50), DVT (I80), PE (I26) and thromboembolic disease (i.e. DVT or PE; I80 or I26). The mortality outcomes included all cause death and cause-specific death, defined based on underlying or contributing causes of death in the NCDR, including cardiovascular death (I00-99) and IHD death (I20-I25).

### Exposure

Covid-19 was defined as a time-varying exposure, meaning that all individuals started as not infected, and they shifted to infected on the date of their first positive PCR test for SARS-CoV-2 during follow-up, using data from the Public Health Agency of Sweden's national register for notifiable communicable diseases (SmiNet). The severity of Covid-19 was assessed, where severe Covid-19 was defined as having been hospitalised due to Covid-19, whereas mild disease was defined as not being hospitalised due to Covid-19. Hospitalisation with Covid-19 was defined as hospitalisation with U07.1 or U07.2 as primary or secondary diagnosis in the NPR, with an admission date within −2 to +14 days from the date of the first positive PCR test.

### Covariates

The sociodemographic data, including age, sex, income, education and country of birth, were obtained from the National Register of the Total Population (RTB) and the Longitudinal Integrated Database for Health Insurance and Labour Market Studies (LISA), at Statistics Sweden. Age was categorised into three groups: 40–54 years, 55–64 years and 65–75 years. Country of birth was categorised as Sweden, high-income countries (HIC) and low- and middle-income countries (LMIC), based on the World Bank classification[35]. The income variable used in this study was the disposable income per consumption unit, and it was calculated by Statistics Sweden by adding the sum of the disposable income of members of the household (i.e., people who are registered in the same home) and then dividing it by the consumption weight that applied to the household[36]. In the analysis, income level was categorised into tertiles (high, medium and low) based on the study cohort. The variable education was categorised as primary schooling (<10 years), secondary schooling (10–12 years) and tertiary schooling (>12 years).

Prior comorbidities, aiming to represent the individual's general health condition before the pandemic started, i.e., markers of their underlying health status relevant to CVD risk, was assessed based on relevant primary or secondary diagnoses registered in the NPR in 2015–2019, and included yes/no variables representing respiratory diseases (ICD-10 codes J00–J99), psychiatric diseases (F20–F39), cancer (C00–C97), diabetes (E10, E11, E13, E14), kidney failure (N17–N19), inflammatory polyarthritis (M05–M14), thyroid disease (E00-E07), coagulation disorder (D65–D69) and obesity (E66).

Data on vaccination against Covid-19 was obtained from the National Vaccination Register (NVR). The first two doses of any Covid-19 vaccinations were considered in the time-varying fashion, meaning that individuals would shift their vaccination status on the date of receiving the first and the second dose.

## Statistical analyses

The descriptive analysis is presented stratified by ever being infected by SARS-CoV-2 or not, *by the end of follow-up*. Mean (standard deviation) is reported for continuous variables and frequency (percentage) for categorical variables. Comparisons between the two groups were performed using a *t*-test for continuous and a Chi-2 test for categorical variables.

Given the nature of the rapid development of the pandemic, the status of Covid-19 and vaccination against Covid-19 were defined as time-varying, e.g., an individual contributed person-time to the "not-infected" group before he/she got Covid-19, and contributed person-time to the "infected" group after Covid-19. An individual's record was further divided according to the date of receiving the first and second doses of vaccination, into periods of not vaccinated, one dose or two doses. Cox proportional hazard models were used to calculate HR and 95% CI[37]. Each individual contributed person-time and outcome event (if any) from the beginning of follow-up (index date) to either the first occurrence of the specific outcome, or censoring due to death, emigration, or end of follow-up, whichever came first. For most of the analyses, three models with different adjustments were used. Model 1 was adjusted for age; Model 2 additionally for sex, country of birth, income, and education; and Model 3 further additionally for comorbidities and vaccination against Covid-19 (fully adjusted model). Furthermore, analyses on the interaction between Covid-19 and vaccination were performed to assess whether the CVD risk was modified by vaccination status.

The risk of CVD and mortality was further analysed by the severity of Covid-19. Moreover, to capture and identify both short- and long-term risk, the risk of CVD and mortality outcomes related to Covid-19 was analysed in a fully adjusted Cox regression model by Covid-19 risk periods defined as 0–14 days, 15–30 days, 31–90 days, 91–180 days, 181–365 days, and 366–730 days, respectively, after infection.

Stratified analyses were further conducted with regard to pandemic wave and predominant virus variants, respectively. The pandemic waves were defined based on definitions by the National Board of Health and Welfare; first wave March 2020-September 2020, second wave October 2020-January 2021, third wave February 2021-June 2021 and fourth wave July 2021-March 2022[38,39]. Predominant virus variants were assessed in line with the time periods corresponding to the dominant virus variant, i.e., mixed variants (January 2020-January 2021), alpha variant (February 2021-June 2021) and delta variant (July 2021-December 2021)[40].

Analyses for three broader key outcomes: cerebrovascular disease, IHD and thromboembolic disease (DVT or PE) were also performed with stratification by sex, age, income, education and country of birth.

Multiple sensitivity analyses were performed. Firstly, sensitivity analyses regarding incident cardiovascular outcomes were made, including data regarding cause-specific deaths corresponding to the studied outcomes from the NCDR, besides diagnoses in the NPR, to capture events that may have resulted in death out of hospital. Secondly, sensitivity analyses were performed among individuals without reported prior comorbidities to address the possibility that the severity of such conditions may influence cardiovascular risk. Thirdly, sensitivity analyses on the age groups 18–39 years and >75 years, respectively, were performed to also address if Covid-19 related cardiovascular risk differs in the excluded younger and older age groups. Fourthly, sensitivity analysis on the risk of cardiovascular disease and mortality outcomes following Covid-19 was assessed, beginning follow-up on 1 March 2020, to include only time with a more clearly documented broad onset of community transmission. Finally, sensitivity analyses excluding those who immigrated during the five-year lookback period were performed since information regarding previous disease among immigrant individuals may be incomplete. All statistics were performed in STATA 18 (StataCorp, TX, USA).

## Reporting summary

Further information on research design is available in the Nature Portfolio Reporting Summary linked to this article.

## Data availability

The data in this study are pseudonymized individual-level data from Swedish healthcare registers and are not publicly available according to Swedish legislation. They can be obtained from the respective Swedish public data holders on the basis of ethics approval for the research in question, subject to relevant legislation, processes and data protection.

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

## Acknowledgements

This work was supported by funding from the SciLifeLab National COVID-19 Research Programme, financed by the Knut and Alice Wallenberg Foundation (grants KAW 2021.0010 [F.N.] and 2020.0299 [F.N.]); the Swedish Research Council (grants 2021-05450 [F.N.] and 2021-05045 [F.N.]); Försäkringskassan (FK 2021/011186 [M.R.]), and the Swedish Heart-Lung Foundation (grants 20210030 [F.N.], 20210581 [F.N.]). The underlying SCIFI-PEARL study also has basic funding by grants from the Swedish state under the agreement between the Swedish government and the county councils, the ALF-agreement (grants ALFGBG-938453 [F.N.], ALFGBG-971130 [F.N.], ALFGBG-978954 [F.N.], ALFGBG-971129 [M.R.]), and ALFGBG-1006729 [F.N.], and previously from a joint grant from Forte (Swedish Research Council for Health, Working Life and Welfare) and FORMAS (Research Council for Environment, Agricultural Sciences and Spatial Planning), (grant 2020-02828 [F.N.]).

## Author contributions

M.S.: contributed to the literature research, conceptualisation, methodology, writing and critical revision of the manuscript. Y.N.D.: contributed to conceptualisation, writing and critical revision of the manuscript. H.L.: contributed to data curation, software, visualisation, data analyses, writing and critical revision of the manuscript. F.N.: Contributed to the investigation, funding acquisition, conceptualisation, methodology, and critical revision of the manuscript. M.R.: contributed to the funding acquisition, conceptualisation, methodology, and critical revision of the manuscript. All co-authors approved the final manuscript.

## Funding

## Competing interests

The authors declare the following financial interests/personal relationships which may be considered as potential competing interests: Dr. Nyberg reports ownership of some Astra Zeneca shares. Dr. Nyberg and Dr. Li report participation in a research projects funded by Bayer and AstraZeneca (regulator-mandated phase IV study; investigator-initiated study), with funds paid to the University of Gothenburg where they are employed (no personal fees) and with no relation to the work reported here. M. Spetz, Dr. Natt och Dag and Dr. Rosvall have nothing to disclose.
