## [Transparent Peer Review file · Nature Communications]

Covid-19 and cardiovascular disease in a total population-study of long-term effects, social factors and Covid-19-vaccination

Corresponding Author: Ms Malin Spetz

Version 0:

Reviewer comments:

Reviewer #1

(Remarks to the Author)

Dear authors,

Thank you for this interesting paper on the association between COVID-19 infection and CVD outcomes. I have several suggestions regarding the methodological framework that could help strengthen the paper's clarity and robustness.

Could you please elaborate on the rationale for restricting analyses to ages 40-75?

The current model controls for age using three broad categories. Given that age is a key determinant of mortality risk, have you considered using age as the time scale in the Cox model rather than adjusting for it as a covariate? This would allow the baseline hazard to vary freely with age, potentially providing more accurate risk estimates.

Regarding the study population definition (Swedish residents as of January 1, 2020, without CVD diagnosis in the preceding five years), could you clarify how migration patterns were handled during the lookback period? Were all individuals observed for the full 5-year period? This is especially important for those born outside of Sweden.

The comorbidity classification would benefit from additional context. What guided the selection of specific categories (e.g., psychiatric diseases)? Was a standardized comorbidity index considered as an alternative or complement to the current approach?

The education variable's categorization uses terms like "mostly" and "generally" - could you provide more precise definitions for these groupings to ensure reproducibility?

Given the prominence of COVID-19 vaccination in the paper's title, would it be valuable to present stratified results for the different vaccination status groups, in addition to using it as a time-varying control variable?

The follow-up period begins January 1st, while the National Board of Health and Welfare defines the first wave's start as March 2020. Have you considered aligning the study period with the documented onset of community transmission in Sweden as done in previous studies?

Reviewer #2

(Remarks to the Author)

Review of Covid-19 and cardiovascular disease: A total population study of long-term effects, social factors and Covid-19-vaccination.

This is a very interesting study that focuses on using a whole population study method for a specific age group (40-75 years of age), and studies a broad range of cardiovascular diseases.

There are some comments to the manuscript though.

Abstract:

- Please be advised that Covid-19 is the disease caused by SARS-CoV-2 infection. Please change "Covid-19 infection" to either just Covid-19 or SARS-CoV-2 infection. Please do this throughout the manuscript.
- It would be clearer to the readers if you define CVDs. You have included venous thromboembolism in this definition.

Methods

- The ICD-10 codes for CVD is rather broad and includes even essential hypertension. I would advise to re-run the analysis excluding only the specific events of interest, and as far back as 1987 from where the inpatient registry is complete. In addition, the Patient Registry only includes specialist healthcare and not primary care, therefore it does not make sense to exclude individuals with any ICD-10 code from I00-I99. An individual could have an I** diagnosis from primary health care and still be included in the analysis.
- What is the rationale for the prior comorbidity ICD-10 codes?
- By removing an individual with a laboratory-verified positive SARS-CoV-2 test from the non-infected population, you are creating a "non-infected" cohort consisting of healthy individuals who may have had Covid-19 but did not test positive. This increases the of outcomes unduly.
- How did you define the starting period of a SARS-CoV-2 infection? Based on what data specifically? In SmiNet there are four potential dates, and not all are complete.

Results:

- In which time period is the HR calculated in Table 2? The risk of events is very high during the first time period, which risks causing significance even when looking at longer time periods.

Reviewer #3

(Remarks to the Author)

In this paper Spetz et al evaluated the risk of cardiovascular disease related to COVID-19 by evaluating all residents of Sweden between January 1st 2020 and December 31st 2021. The authors modeled COVID-19 as a time-varying exposure and identified higher risk for acute myocardial infarction and pulmonary embolism, particularly for participants who were hospitalized. The following are some comments regarding methods, statistical approaches and reporting:

1. Please report the overall number of cardiovascular events recorded in the cohort during the study period in the abstract.
2. Using hospitalization as a proxy of disease severity is questionable, especially if hospital occupancy was modified during pandemic peaks and thus hospitalization criteria changed over time. As there is likely no reliable alternative for severity over time, the limitations of using this approach should be discussed at length.
3. A main concern of attributing risk of cardiovascular mortality due to COVID-19 may come from reverse causality. Individuals with comorbidities are likely already at a higher risk of CVD compared to those without comorbidity and the onset of COVID-19 may lead to decompensation of comorbidities rather than COVID-19 related risk. Therefore, authors should perform a sensitivity analysis in individuals without reported comorbidities to assess whether the effect of reverse causality can be minimized. This concern is not fully addressed by adjustment, as this does not inherently address the possibility of increased disease severity for comorbidities underlying CVD risk. Please include this analysis and discuss its potential implications.
4. COVID-19 vaccination is treated as an adjustment variable only in model 3, but this should be accounted for even in the minimally adjusted model. Authors should include vaccination status as a relevant covariate in all models to account for its effect on CVD risk.
5. Furthermore, the impact of vaccination should be explicitly modeled by including a multiplicative or additive interaction with COVID-19 status to evaluate its effect modification in CVD risk. A subgroup analysis by number of vaccine doses should also be considered to assess whether the risk was modified by this factor. Furthermore, effectiveness of COVID-19 vaccines as protective for risk of CVD after COVID-19 could also be derived from Cox models and would be a relevant addition to the analysis.
6. Please clarify why only four disease categories were considered for adjustment. Were other comorbidities excluded for a specific reason? How were individuals with other diagnoses accounted for in adjustment? A clear rationale should be provided for this.
7. Is there a particular reason why only individuals within 40-75 years considered? CVD risk can be increased for individuals above and below this range and no clear rationale is provided for only analyzing this age range. How does COVID-19 related CVD risk change in younger and older adults? If this data is available it should be provided and can then be analyzed separately if needed, but it should be included and reported.
8. Given that this is a whole population cohort, all incident CVD outcomes should be reported in incidence rates with their corresponding uncertainty measures (to account for potential unidentified incident cases). The full person-years of follow-up for all participants should be reported in the main results section.
9. Considering that data on major circulating SARS-CoV-2 variants per date is available, it would be useful to consider them in the model by including the predominant variant per each time period as both an adjustment covariate and as a stratification or interaction term. This would be a unique opportunity to report whether distinct predominating SARS-CoV-2 variants had distinct impacts on the risk for CVD outcomes after COVID-19.
10. Is reinfection data available in this cohort? It would be interesting to explore repeated events and its impact on CVD risk, particularly for those who only developed CVD after the second exposure to COVID-19. Given evidence that prior infection likely provided protection for severe disease in subsequent infections, decreased CVD risk could be hypothesized. If this

data is available it would be useful to analyze it. Otherwise, this should be discussed, especially as reinfection is a very relevant concern for SARS-CoV-2 infections after the pandemic.

Version 1:

Reviewer comments:

Reviewer #1

(Remarks to the Author)

The authors made meaningful improvements to the manuscript. I have a few remaining comments regarding the authors replies.

The authors wrote "Firstly, although presented as three categories in the descriptives, age was adjusted as a continuous variable in the models. Secondly, given that Covid-19 rapidly evolved during the pandemic, as evidenced by waves and variants of concerns, it may be advantageous, or even necessary, to use calendar time scale to avoid potential biases."

I understand the reasoning. Two points: First, the time scale should be explicitly mentioned when Cox regression is introduced. Second, this also implies that the effect of all covariates is assumed to be proportional across calendar time. Given the evolution of the COVID-19 pandemic across the two years, this is a strong assumption that could lead to biased estimates if treatment effects or other covariate relationships changed between waves or variants. While your separate models by pandemic wave help address this proportionality issue, the wave-specific covariate effects are not presented to readers, making it difficult to assess whether important temporal changes occurred. I suggest either reporting these wave-specific effects or explicitly discussing this methodological consideration in the limitations.

The authors wrote "Since it is not appropriate to stratify on a variable (or status) that is time-varying, we have instead performed analyses on the interaction between vaccination status and Covid-19 on CVD risk."

It is possible and appropriate to stratify on a time-varying covariate in Cox regression using the strata() option in Stata after the data is stsplit.

The authors wrote "Since we use a time-varying variable for Covid-19, there is no evident problem of starting follow-up on January 1st 2020 even though there are very few cases during January and February 2020." in response to my comment "The follow-up period begins January 1st, while the National Board of Health and Welfare defines the first wave's start as March 2020. Have you considered aligning the study period with the documented onset of community transmission in Sweden as done in previous studies?"

I disagree. I assume you refer to the time-varying variable for individual COVID-19 infection. As testing was unavailable in the first months of 2020 and transmission was low any observations contributing events and exposure time to this period would not be of interest to the research question. This pre-transmission period essentially adds noise rather than meaningful signal to your analysis. The time-varying approach you mention doesn't resolve the fundamental issue that you're including a period where the exposure of interest (COVID-19 transmission) was essentially absent.

Furthermore, including January-February 2020 as follow-up time when COVID-19 transmission was negligible could bias your effect estimates by diluting the true association between COVID-19 waves and your outcome. The methodological precedent of aligning study periods with documented transmission onset exists precisely to avoid this issue.

Reviewer #3

(Remarks to the Author)

Thank you for all your revisions and for all additional analyses, they have greatly strengthened the results and improved your manuscript. I only have one additional query:

- Regarding vaccination analyses in Supplementary Figure 1, authors identified a higher risk of ischemic heart disease for participants without infection but with prior vaccination compared to those unvaccinated without infection. Can the authors further comment on these findings? Were there any significant differences between those unvaccinated, those vaccinated with 1 and 2 doses? Differences across these subgroups may explain differential risks, which could be contrasted with previous findings (Please see: 10.1001/jama.2022.1299, 10.4103/ijpvm.ijpvm_260_24). Do the authors know which vaccines were used during this period? This could also align with some additional findings regarding risk of ischemic heart disease with specific vaccine types (10.4103/ijpvm.ijpvm_260_24).

REVIEWER COMMENTS

Reviewer 1

Rev: 1. Thank you for this interesting paper on the association between COVID-19 infection and CVD outcomes. I have several suggestions regarding the methodological framework that could help strengthen the paper's clarity and robustness.

Answer: Thank you!

Rev: 2. Could you please elaborate on the rationale for restricting analyses to ages 40-75?

Answer: Thank you for raising this important consideration. In our study, we address individuals aged 40-75 years. The rationale for the lower age limit was based on low incidence rates of the diagnoses studied in younger age, potentially leading to problems with low power (*please see references 31 and 32 in the manuscript*), in combination with a different pathophysiology, where genetic and autoimmune disorders play a relatively more prominent role (*please see reference 30 in the manuscript*). Furthermore, the rationale behind the higher age limit was based on the fact that in the beginning of the pandemic, when the test capacity for Covid-19 was limited, older individuals staying at retirement homes in Sweden were tested to a lower extent (*please see reference 33 in the manuscript*). The rationale for restricting the analyses to the ages 40-75 is now described in the Method section on page 13 paragraph 3.

However, to ensure the comprehensiveness of our study, we have now also performed additional sensitivity analyses (described in the Methods section page 16, paragraph 5 and page 17, paragraph 1) on the age groups 18-39 years and > 75 years, respectively, to address also Covid-19 related cardiovascular risk and mortality in younger (Supplementary Table 4 and Supplementary Table 5) and older (Supplementary Table 6 and Supplementary Table 7) age groups. The results are described in the Results section on page 5, paragraph 2.

Rev: 3. The current model controls for age using three broad categories. Given that age is a key determinant of mortality risk, have you considered using age as the time scale in the Cox model rather than adjusting for it as a covariate? This would allow the baseline hazard to vary freely with age, potentially providing more accurate risk estimates.

Answer: Thank you for a valuable comment. Firstly, although presented as three categories in the descriptives, age was adjusted as a continuous variable in the models. Secondly, given that Covid-19 rapidly evolved during the pandemic, as evidenced by waves and variants of concerns, it may be advantageous, or even necessary, to use calendar time scale to avoid potential biases.

Rev: 4. Regarding the study population definition (Swedish residents as of January 1, 2020, without CVD diagnosis in the preceding five years), could you clarify how migration patterns were handled during the lookback period? Were all individuals observed for the full 5-year period? This is especially important for those born outside of Sweden.

Answer: Thank you for this comment. We have now performed sensitivity analyses (described in the Methods section page 17, paragraph 1) excluding those who immigrated during the five-year

lookback period regarding the main analyses (i.e., Table 2, Supplementary Table 1 and Table 3) and the results showed similar patterns of associations (data not shown) described in the Results section on page 5, paragraph 2, page 6, paragraph 1 and on page 8, paragraph 1.

Rev: 5. The comorbidity classification would benefit from additional context. What guided the selection of specific categories (e.g., psychiatric diseases)? Was a standardized comorbidity index considered as an alternative or complement to the current approach?

Answer: Thank you for this important question. The comorbidities selected for adjustment in this study are diagnoses aimed to represent the individual's general health condition before the pandemic started, i.e., markers of their underlying health status relevant to CVD risk. We have now added additional diagnosis groups known to be connected to cardiovascular risk to even better describe such baseline health status i.e., Kidney failure (N17-N19), Inflammatory polyarthropathies (M05-M14), Thyroid disease (E00-E07), Coagulation disorder (D65-D69) and Obesity (E66).

We chose not to use a comorbidity index since we were not able to find a standardised comorbidity index available for Covid-19 related complications. For example, the Charlson Comorbidity Index (CCI), often used as a proxy for comorbidity burden, was originally developed to predict short-time mortality and thus only encompasses comorbidities associated with such increased mortality risk, and, the CCI has for example often been used in cancer research.

Rev: 6. The education variable's categorization uses terms like "mostly" and "generally" - could you provide more precise definitions for these groupings to ensure reproducibility?

Answer: Thank you for an important comment. We have now given more precise definitions on the educational variables in the Method section on page 15, paragraph 1.

Rev: 7. Given the prominence of COVID-19 vaccination in the paper's title, would it be valuable to present stratified results for the different vaccination status groups, in addition to using it as a time-varying control variable?

Answer: Thank you for pointing this out. Since it is not appropriate to stratify on a variable (or status) that is time-varying, we have instead performed analyses on the interaction between vaccination status and Covid-19 on CVD risk. The results are now presented in Supplementary Figure 1 and described in the Results section on page 5, paragraph 1.

Rev: 8. The follow-up period begins January 1st, while the National Board of Health and Welfare defines the first wave's start as March 2020. Have you considered aligning the study period with the documented onset of community transmission in Sweden as done in previous studies?

Answer: Thanks for your valuable comment. Since we use a time-varying variable for Covid-19, there is no evident problem of starting follow-up on January 1st 2020 even though there are very few cases during January and February 2020.

Reviewer #2 (Remarks to the Author):

Review of Covid-19 and cardiovascular disease: A total population study of long-term effects, social factors and Covid-19-vaccination.

Rev: 1. This is a very interesting study that focuses on using a whole population study method for a specific age group (40-75 years of age), and studies a broad range of cardiovascular diseases.

There are some comments to the manuscript though.

Answer: Thank you, please see our answers to the comments below.

Abstract:

Rev: 2. - Please be advised that Covid-19 is the disease caused by SARS-CoV-2 infection. Please change "Covid-19 infection" to either just Covid-19 or SARS-CoV-2 infection. Please do this throughout the manuscript.

Answer: Thanks for pointing this out, we have now made suggested changes throughout the manuscript.

Rev: 3. - It would be clearer to the readers if you define CVDs. You have included venous thromboembolism in this definition.

Answer: Thank you for this important comment. We have now made clarifications in the manuscript regarding the definition of cardiovascular diseases according to the WHO, please see Introduction section, page 3, paragraph 1 (reference 4 in the manuscript).

Methods

Rev: 4. - The ICD-10 codes for CVD is rather broad and includes even essential hypertension. I would advise to re-run the analysis excluding only the specific events of interest, and as far back as 1987 from where the inpatient registry is complete. In addition, the Patient Registry only includes specialist healthcare and not primary care, therefore it does not make sense to exclude individuals with any ICD-10 code from I00-I99. An individual could have an I** diagnosis from primary health care and still be included in the analysis

Answer: Thank you for this really important comment. According to your suggestion, we now only exclude individuals with prior cardiovascular events corresponding to the studied outcomes during five years before index date. Based on our research topic and ethics approvals, the available dataset only includes data as far back as 2015 so unfortunately, we cannot go further back than that. However, these diseases are chronic diseases that would be expected to be fairly well captured over a 5-year period if they affect the health status of the individual significantly.

Rev: 5. - What is the rationale for the prior comorbidity ICD-10 codes?

Answer: Thank you for this important question. The comorbidities selected for adjustment in this study are diagnoses aimed to represent the individual's general health condition before the pandemic started, i.e. markers of their underlying health status relevant to CVD risk. We have now added additional diagnoses groups known to be connected to cardiovascular risk to even better describe such baseline health status i.e., Kidney failure (N17-N19), Inflammatory polyarthropaties (M05-M14), Thyroid disease (E00-E07), Coagulation disorder (D65-D69) and Obesity (E66).

We chose not to use a comorbidity index since we were not able to find a standardized comorbidity index available for Covid-19 related complications. For example, the Charlson Comorbidity Index (CCI), often used as a proxy for comorbidity burden, was originally developed to predict short-time mortality and thus only encompasses comorbidities associated with such increased mortality risk and, the CCI has for example often been used in cancer research.

Rev: 6. - By removing an individual with a laboratory-verified positive SARS-CoV-2 test from the non-infected population, you are creating a "non-infected" cohort consisting of healthy individuals who may have had Covid-19 but did not test positive. This increases the of outcomes unduly.

Answer: Thank you for your valuable comment. The fact that, especially during the early pandemic, the test capacity for SARS-CoV-2 was limited and individuals with mild Covid-19 were not prioritized, was previously discussed as a limitation. However, we have now also added a sentence in the Discussion section (page 12, paragraph 2) discussing that the restricted testing implying that some individuals with Covid-19 were classified as non-infected could underestimate the demonstrated risks for CVD. In addition, sensitivity analyses showed similar patterns of associations across pandemic phases, although with stronger associations during the first and second pandemic phases, indicating robustness of findings despite variation in SARS-CoV-2 test capacity.

Rev: 7. - How did you define the starting period of a SARS-CoV-2 infection? Based on what data specifically? In SmiNet there are four potential dates, and not all are complete.

Answer: Thank you for this specific question. Using the data from SmiNet, we defined the date of SARS-CoV-2 infection as "Provtagningsdatum_fall" when available. Otherwise, "Statistikdatum" is used. "Statistikdatum" has no missing in SmiNet data.

Results:

Rev: 8. - In which time period is the HR calculated in Table 2? The risk of events is very high during the first time period, which risks causing significance even when looking at longer time periods.

Answer: Thank you for this comment. In Table 2, the overall risks (HRs) for cardiovascular disease from 1 January 2020 to 31 December 2021 are demonstrated. However, to better illustrate risks of cardiovascular disease in different risk periods we show this in Figure 3 and Supplementary Table 12 and described in the Results section on page 7, paragraph 2.

Reviewer #3 (Remarks to the Author):

In this paper Spetz et al evaluated the risk of cardiovascular disease related to COVID-19 by evaluating all residents of Sweden between January 1st 2020 and December 31st 2021. The authors modeled COVID-19 as a time-varying exposure and identified higher risk for acute myocardial infarction and pulmonary embolism, particularly for participants who were hospitalized. The following are some comments regarding methods, statistical approaches and reporting:

Rev: 1. Please report the overall number of cardiovascular events recorded in the cohort during the study period in the abstract.

Answer: Thank you for this comment. However, since the cardiovascular outcomes may overlap, e.g., the same individual can be in both the AMI and ischemic heart disease outcome group, it is hard to simply add up each outcome. One solution would be to mention the larger outcome groups, e.g., cerebrovascular disease, ischemic heart disease, thromboembolic disease etc., however, since there is a strong restriction regarding number of words, we suggest that we do not present overall numbers in the Abstract, but of course we present the number of events for each outcome including incidence rates, in Table 2 and Supplementary Table 1. We hope the reviewer finds this ok.

Rev: 2. Using hospitalization as a proxy of disease severity is questionable, especially if hospital occupancy was modified during pandemic peaks and thus hospitalization criteria changed over time. As there is likely no reliable alternative for severity over time, the limitations of using this approach should be discussed at length.

Answer: Thank you for this important comment. We now discuss the limitations of using hospitalisation as a proxy of disease severity over time, please see Discussion section, page 12, paragraph 2.

Rev: 3. A main concern of attributing risk of cardiovascular mortality due to COVID-19 may come from reverse causality. Individuals with comorbidities are likely already at a higher risk of CVD compared to those without comorbidity and the onset of COVID-19 may lead to decompensation of comorbidities rather than COVID-19 related risk. Therefore, authors should perform a sensitivity analysis in individuals without reported comorbidities to assess whether the effect of reverse causality can be minimized. This concern is not fully addressed by adjustment, as this does not inherently address the possibility of increased disease severity for comorbidities underlying CVD risk. Please include this analysis and discuss its potential implications.

Answer: Thank you for this valuable comment. We have now performed sensitivity analyses among individuals without reported prior comorbidities to address the possibility that the severity of such conditions may influence cardiovascular risk. The results are presented in the Results section on page 5, paragraph 2, (and shown in Supplementary Table 2 and Supplementary Table 3), on page 7, paragraph 3 and on page 8, paragraph 1, (and shown in Supplementary Table 13).

Rev: 4. COVID-19 vaccination is treated as an adjustment variable only in model 3, but this should be accounted for even in the minimally adjusted model. Authors should include vaccination status as a relevant covariate in all models to account for its effect on CVD risk.

Answer: Thank you for this valuable comment. We have now chosen to consistently present model 3, which is the most complete model, throughout the manuscript.

Rev: 5. Furthermore, the impact of vaccination should be explicitly modeled by including a multiplicative or additive interaction with COVID-19 status to evaluate its effect modification in CVD risk. A subgroup analysis by number of vaccine doses should also be considered to assess whether the risk was modified by this factor. Furthermore, effectiveness of COVID-19 vaccines as protective for risk of CVD after COVID-19 could also be derived from Cox models and would be a relevant addition to the analysis.

Answer: Thanks for a relevant comment. Since it is not appropriate to stratify on a variable (or status) that is time-varying, we have instead performed analyses on the interaction between the vaccination status (non-vaccinated, 1 dose, 2 doses or more) and Covid-19 on CVD risk. The results are now presented in the Results section on page 5, paragraph 1 and shown in Supplementary Figure 1.

Rev: 6. Please clarify why only four disease categories were considered for adjustment. Were other comorbidities excluded for a specific reason? How were individuals with other diagnoses accounted for in adjustment? A clear rationale should be provided for this.

Answer: Thank you for this important question. The comorbidities selected for adjustment in this study are diagnoses aimed to represent the individual's general health condition before the pandemic started, i.e. markers of their underlying health status relevant to CVD risk. We have now added additional diagnoses groups known to be connected to cardiovascular risk to even better describe such baseline health status i.e., Kidney failure (N17-N19), Inflammatory polyarthropaties (M05-M14), Thyroid disease (E00-E07), Coagulation disorder (D65-D69) and Obesity (E66).

We chose not to use a comorbidity index since we were not able to find a standardized comorbidity index available for Covid-19 related complications. For example, the Charlson Comorbidity Index (CCI), often used as a proxy for comorbidity burden, was originally developed to predict short-time mortality and thus only encompasses comorbidities associated with such increased mortality risk and, the CCI has for example often been used in cancer research.

Rev: 7. Is there a particular reason why only individuals within 40-75 years considered? CVD risk can be increased for individuals above and below this range and no clear rationale is provided for only analyzing this age range. How does COVID-19 related CVD risk change in younger and older adults? If this data is available it should be provided and can then be analyzed separately if needed, but it should be included and reported.

Answer: Thank you for raising this important consideration. In our study, we address individuals aged 40-75 years. The rationale for the lower age limit was based on low incidence rates of the diagnoses studied in younger age, potentially leading to problems with low power (*please see*

references 31 and 32 in the manuscript), in combination with a different pathophysiology, where genetic and autoimmune disorders play a relatively more prominent role (*please see reference 30 in the manuscript*). Furthermore, the rationale behind the higher age limit was based on the fact that in the beginning of the pandemic, when the test capacity for Covid-19 was limited, older individuals staying at retirement homes in Sweden were tested to a lower extent (*please see reference 33 in the manuscript*). The rationale for restricting the analyses to the ages 40-75 is now described in the Method section on page 13 paragraph 3.

However, to ensure the comprehensiveness of our study, we have now also performed additional sensitivity analyses (described in the Methods section page 16, paragraph 5 and page 17, paragraph 1) on the age groups 18-39 years and > 75 years, respectively, to address also Covid-19 related cardiovascular risk and mortality in younger (Supplementary Table 4 and Supplementary Table 5) and older (Supplementary Table 6 and Supplementary Table 7) age groups. The results are described in the Results section on page 5, paragraph 2.

Rev: 8. Given that this is a whole population cohort, all incident CVD outcomes should be reported in incidence rates with their corresponding uncertainty measures (to account for potential unidentified incident cases). The full person-years of follow-up for all participants should be reported in the main results section.

Answer: Thank you for this comment. We have now added incidence rates with corresponding uncertainty measures in Table 2 and Supplementary Table 1 and the full person-years of follow-up for all participants in the main Results section on page 5, paragraph 1.

Rev: 9. Considering that data on major circulating SARS-CoV-2 variants per date is available, it would be useful to consider them in the model by including the predominant variant per each time period as both an adjustment covariate and as a stratification or interaction term. This would be a unique opportunity to report whether distinct predominating SARS-CoV-2 variants had distinct impacts on the risk for CVD outcomes after COVID-19.

Answer: Thank you for your valuable comment. We have now performed stratified analyses investigating predominant virus variants, please see Results section on page 7, paragraph 1 (Supplementary Table 11).

Rev: 10. Is reinfection data available in this cohort? It would be interesting to explore repeated events and its impact on CVD risk, particularly for those who only developed CVD after the second exposure to COVID-19. Given evidence that prior infection likely provided protection for severe disease in subsequent infections, decreased CVD risk could be hypothesized. If this data is available it would be useful to analyze it. Otherwise, this should be discussed, especially as reinfection is a very relevant concern for SARS-CoV-2 infections after the pandemic.

Answer: Thank you for an important comment. Unfortunately, multiple infections with Covid-19 was not considered in our analyses since information on reinfection was limited in the Swedish Covid-19 surveillance. This is now mentioned as a limitation in the Discussion section on page 12, paragraph 2.

REVIEWER COMMENTS

Reviewer #1 (Remarks to the Author):

Rev: The authors made meaningful improvements to the manuscript.

Answer: Thank you!

Rev: I have a few remaining comments regarding the authors replies.

Rev: The authors wrote "Firstly, although presented as three categories in the descriptives, age was adjusted as a continuous variable in the models. Secondly, given that Covid-19 rapidly evolved during the pandemic, as evidenced by waves and variants of concerns, it may be advantageous, or even necessary, to use calendar time scale to avoid potential biases."

I understand the reasoning. Two points: First, the time scale should be explicitly mentioned when Cox regression is introduced. Second, this also implies that the effect of all covariates is assumed to be proportional across calendar time. Given the evolution of the COVID-19 pandemic across the two years, this is a strong assumption that could lead to biased estimates if treatment effects or other covariate relationships changed between waves or variants. While your separate models by pandemic wave help address this proportionality issue, the wave-specific covariate effects are not presented to readers, making it difficult to assess whether important temporal changes occurred. I suggest either reporting these wave-specific effects or explicitly discussing this methodological consideration in the limitations.

Answer: Thank you for important additional comments regarding the time scale. We have now also added in the Limitation section (Please see page 11, paragraph 3) that "An additional limitation is that in order to interpret a summary HR as valid throughout follow-up, all covariates included in the Cox model are assumed to have more or less proportional effects across time, which may not always be the case. Yet our wave-specific analysis provided consistent patterns of associations which potentially indicates that this issue is not a substantial concern." Furthermore, we note that even with hazards that do vary over follow-up (as commonly occurs), the HR is reasonably interpreted as a time-weighted average effect, as others have shown.

Rev: The authors wrote "Since it is not appropriate to stratify on a variable (or status) that is time-varying, we have instead performed analyses on the interaction between vaccination status and Covid-19 on CVD risk."

It is possible and appropriate to stratify on a time-varying covariate in Cox regression using the `strata()` option in Stata after the data is `stsplit`.

Answer: Thank you for this valuable comment. We have now performed a stratified Cox regression using the STATA option `strata()` in `stcox` as suggested by the reviewer, to investigate the overall combined effect of infection (on Table 2 and Supplementary table 1) – among all possible strata of vaccination status allowing for variable baseline hazards in the different strata. The results (see Table below, rightmost column) were very similar to the results presented in

Table 2 and Supplementary Table 1 where vaccination was adjusted as a time-varying variable (Table below, "Model 3" column).

Incident CVD outcomes and mortality	Covid-19 ^c	Model 3 ^a	Fully adjusted but with Cox-stratification for vaccination status ^b
		HR (95% CI)	HR (95% CI)
Ischemic stroke	No	1.00 ^e	1.00 ^e
	Yes	1.38 (1.27-1.48)	1.37 (1.27-1.48)
Intracerebral hemorrhage	No	1.00 ^e	1.00 ^e
	Yes	1.44 (1.23-1.69)	1.43 (1.22-1.67)
Cerebrovascular disease	No	1.00 ^e	1.00 ^e
	Yes	1.45 (1.36-1.53)	1.43 (1.35-1.52)
Acute myocardial infarction	No	1.00 ^e	1.00 ^e
	Yes	1.22 (1.14-1.31)	1.22 (1.14-1.31)
Ischemic heart disease	No	1.00 ^e	1.00 ^e
	Yes	1.35 (1.28-1.41)	1.34 (1.28-1.41)
Cardiomyopathy	No	1.00 ^e	1.00 ^e
	Yes	1.39 (1.21-1.61)	1.39 (1.20-1.60)
Heart failure	No	1.00 ^e	1.00 ^e
	Yes	1.54 (1.45-1.64)	1.52 (1.43-1.62)
Deep venous thrombosis	No	1.00 ^e	1.00 ^e
	Yes	1.77 (1.66-1.89)	1.77 (1.65-1.88)
Pulmonary embolism	No	1.00 ^e	1.00 ^e
	Yes	4.31 (4.09-4.55)	4.27 (4.05-4.50)
All-cause mortality	No	1.00 ^e	1.00 ^e
	Yes	3.19 (3.08-3.30)	3.02 (2.92-3.12)
Cardiovascular mortality	No	1.00 ^e	1.00 ^e
	Yes	3.46 (3.28-3.65)	3.28 (3.11-3.46)
Ischemic heart disease mortality	No	1.00 ^e	1.00 ^e
	Yes	1.76 (1.56-1.99)	1.71 (1.51-1.93)

- Model 3: Adjusted for age, sex, country of birth, income, education, comorbidities and vaccination against Covid-19
- Stratified Model 3: Adjusted for age, sex, country of birth, income, education, and comorbidities, stratified for vaccination against Covid-19

Rev: The authors wrote "Since we use a time-varying variable for Covid-19, there is no evident problem of starting follow-up on January 1st 2020 even though there are very few cases during January and February 2020." in response to my comment "The follow-up period begins January 1st, while the National Board of Health and Welfare defines the first wave's start as March 2020. Have you considered aligning the study period with the documented onset of community transmission in Sweden as done in previous studies?"

I disagree. I assume you refer to the time-varying variable for individual COVID-19 infection. As

testing was unavailable in the first months of 2020 and transmission was low any observations contributing events and exposure time to this period would not be of interest to the research question. This pre-transmission period essentially adds noise rather than meaningful signal to your analysis. The time-varying approach you mention doesn't resolve the fundamental issue that you're including a period where the exposure of interest (COVID-19 transmission) was essentially absent.

Furthermore, including January-February 2020 as follow-up time when COVID-19 transmission was negligible could bias your effect estimates by diluting the true association between COVID-19 waves and your outcome. The methodological precedent of aligning study periods with documented transmission onset exists precisely to avoid this issue.

Answer: Thank you for a valuable comment. The first known (identified) Covid-19 positive case was reported in January 2020 in Sweden and a number of cases (approximately 5-10 cases) were reported in February 2020 in Sweden. In order to include all Covid-19 infection instances, we still think it makes sense to start the follow up from 1 January 2020. Moreover, the low number of cases per se does not invalidate the analysis, nor dilute it, since the comparison is only performed for risk sets on each day and thus valid, and the time-weighted HR receives very little or no weight from this portion of follow-up if there are no cases. Nevertheless, in order to validate our results, we have now re-run the analysis on the risk of cardiovascular disease outcomes following Covid-19 (i.e., Table 2, model 3) with starting follow-up 1 March 2020 and the results are essentially identical. For example, the HRs with 95% CI for model 3 were as follows, please see table beneath. However, we have now added comments on this in the Results section (please see page 5, paragraph 2), in the Limitation section (please see page 12, paragraph 1), and in the Method section (please see page 16, paragraph 3), respectively.

Incident CVD outcomes	Covid-19	Starting follow-up 1 Jan 2020	Starting follow-up 1 Mar 2020
		HR (95% CI)	HR (95% CI)
	No	1.00	1.00
	Yes	1.38 (1.27-1.48)	1.38 (1.27-1.48)
Ischemic stroke	No	1.00	1.00
	Yes	1.44 (1.23-1.69)	1.44 (1.23-1.69)
Intracerebral hemorrhage	No	1.00	1.00
	Yes	1.45 (1.36-1.53)	1.44 (1.36-1.53)
Cerebrovascular disease	No	1.00	1.00
	Yes	1.22 (1.14-1.31)	1.22 (1.14-1.31)
Acute myocardial infarction	No	1.00	1.00
	Yes	1.35 (1.28-1.41)	1.34 (1.28-1.41)
Ischemic heart disease	No	1.00	1.00
	Yes	1.39 (1.21-1.61)	1.39 (1.20-1.60)
Cardiomyopathy	No	1.00	1.00
	Yes	1.54 (1.45-1.64)	1.54 (1.45-1.64)
Heart failure	No	1.00	1.00
	Yes	1.00	1.00
Deep venous thrombosis	No	1.00	1.00

	Yes	1.77 (1.66-1.89)	1.77 (1.66-1.89)
	No	1.00	1.00
Pulmonary embolism	No	1.00	1.00
	Yes	4.31 (4.09-4.55)	4.32 (4.10-4.56)
	No	1.00	1.00
All cause mortality	No	1.00	1.00
	Yes	3.19 (3.08-3.30)	3.21 (3.10-3.32)
	No	1.00	1.00
Cardiovascular disease mortality	No	1.00	1.00
	Yes	3.46 (3.28-3.65)	3.47 (3.29-3.66)
	No	1.00	1.00
Ischemic heart disease mortality	No	1.00	1.00
	Yes	1.76 (1.56-1.99)	1.77 (1.56-2.00)

Reviewer #3 (Remarks to the Author):

Rev: Thank you for all your revisions and for all additional analyses, they have greatly strengthened the results and improved your manuscript.

Answer: Thank you!

Rev: I only have one additional query:

- Regarding vaccination analyses in Supplementary Figure 1, authors identified a higher risk of ischemic heart disease for participants without infection but with prior vaccination compared to those unvaccinated without infection. Can the authors further comment on these findings? Were there any significant differences between those unvaccinated, those vaccinated with 1 and 2 doses? Differences across these subgroups may explain differential risks, which should be contrasted with previous findings (Please see: 10.1001/jama.2022.1299, 10.4103/ijpvm.ijpvm_260_24). Do the authors know which vaccines were used during this period? This could also align with some additional findings regarding risk of ischemic heart disease with specific vaccine types (10.4103/ijpvm.ijpvm_260_24).

Answer: Thank you for this comment. The reviewer is correct that there is a slightly increased risk of ischemic heart disease for participants without infection but with prior vaccination, however, among those infected the results instead indicate a cardioprotective role of prior vaccination on the risk of IHD. In our view, the point estimate of 1.08-1.09 is so close to 1 that the CI:s, even if not including 1 (due to the large sample size), cannot be strongly relied on. Furthermore, the risk of MI (the most serious kind of IHD) is not elevated. Given the different estimates produced in the study, some variability can be expected, and the pattern for IHD may be reasonably interpreted as no clear effect trend. We would prefer not to further specifically comment on the very slightly increased risk of IHD among participants without infection but with prior vaccination in the paper since we do not think it is connected to a clinically relevant risk increase.

A previous study, also using data from the SCIFI-PEARL, but investigating cardiovascular events after Covid-19 vaccination, generally showed that the risk for various cardiac outcomes, among those aged >40 years, decreased after Covid-19 vaccination, even though there was a transiently increased risk for extrasystoles. These findings are described and referred to in the Discussion section (ref 26) on page 10, paragraph 3. In the referred paper separate analyses on the three vaccine products mainly used in Sweden: BNT162b2 (Pfizer-BioNTech), mRNA-1273

(Moderna), and AZD1222 (AstraZeneca) were made showing no difference in cardiac risk between vaccine products. We have now added also a description of the findings connected to the separate analyses of the three vaccine products.